# Design, Analysis and Experiments of Hexapod Robot with Six-Link Legs for High Dynamic Locomotion

**DOI:** 10.3390/mi13091404

**Published:** 2022-08-26

**Authors:** Jiawang Ma, Guanlin Qiu, Weichen Guo, Peitong Li, Gan Ma

**Affiliations:** Sino-German College of Intelligent Manufacturing, Shenzhen Technology University, Shenzhen 518118, China

**Keywords:** hexapod robot, six-link mechanism, high dynamic capability

## Abstract

An important feature of a legged robot is its dynamic motion performance. Traditional methods often improve the dynamic motion performance by reducing the moment of inertia of robot legs or by adopting quasi-direct drive actuators. This paper proposes a method to enhance the dynamic performance of a legged robot by transmission mechanism. Specifically, we present a unique six-link leg mechanism that can implement a large output motion using a small drive motion. This unique feature can enhance the robots’ dynamic motion capability. Experiments with a hexapod robot verified the effectiveness of the mechanism. The experimental results showed that, when the steering gear of the robot rotates 1°, the toe can lift 7 mm (5% of body height), and the maximum running speed of the robot can reach 390 mm/s (130% of the moveable body length per second).

## 1. Introduction

Legged robots have better flexibility and mobility than wheeled or tracked vehicles in uneven terrain such as uplands, forests, and stairways. Research shows that most existing quadruped robots have high maneuverability, highly dynamic, and agile locomotion [1,2,3,4,5,6,7,8,9], while hexapod robots have high stability to traverse unstructured terrain [10,11]. To improve the dynamic motion capability of a hexapod robot, it is desirable to improve its transmission mechanism [12,13].

As the main component of a multi-legged robot, the leg mechanism plays the role of achieving motion, weight-bearing, and balancing. Therefore, the study of leg mechanisms is one of the core parts in research on multi-legged robots. Compared with quadruped robots, hexapod robots have higher terrain adaptability but lose some speed. In some flat terrains where stability is not required, an important index affecting the efficiency of a hexapod robot is its dynamic motion capability. For hexapod robots, the traditional methods to improve the dynamic motion capability of the legs are divided into the following: the cable-driven method [14,15,16,17] and the one quasi-direct drive actuator method [18]. The special structure of the cable-driven gives it a low inertia. The advantages of the cable-driven method are the simple structure, low cost, and low noise. However, the basis of the cable-driven method is friction, so this method has low drive efficiency and unstable drive. In addition, this method has low life and low payload. The one quasi-direct drive actuator method uses a high torque density motor with a low-speed ratio reducer, coupled with a low inertia leg design. The advantages include its compactness, the ability to achieve high bandwidth force control, and good shock resistance. However, quasi-direct drive actuators tend to be expensive.

Therefore, we propose a six-link leg mechanism that can significantly improve the dynamic motion of a hexapod robot by relying only on mechanical structures. This mechanism has the following characteristics:This mechanism can realize a transformation from small input motion to large output motion.Comparing the two methods mentioned above, the method using this mechanism has stable transmission, low wear, large payload, and low cost.This method has the advantages of easy processing, reliable working member, and easy lubrication.

The hexapod robot proposed in this paper uses this leg mechanism to ensure high dynamic performance of the hexapod robot by carrying inexpensive servos in each leg only, which greatly reduces the production cost.

In motion planning, conventional hexapod robots usually use tripod [19,20,21,22,23], quadripod [24], pentapod gaits (Fluctuating gait) [25,26,27], or hexapod gait [28,29] for walking. Quadruped gait means that the robot always has four legs in a supported phase during motion. Pentapod gait, also called fluctuating gait, is a gait in which the robot has only one leg swinging at all times during its motion. These gaits satisfy the hydrostatic conditions and have strong terrain adaptability. However, when the hexapod robot walks on the ground with low stability requirements, it may lose a lot of speed. Therefore, based on the stability and speed considerations of the robot, this paper introduces a tripod gait and a bipod gait to apply different motion scenarios, just like the gait transition rules proposed by Yasushi Habu et al. [30]. The research in [31] developed a general method to learn from the morphology the appropriate coupling weights between sensory feedback and the limb phase oscillators to form an adaptive locomotion controller. In general terms, this controller can switch the robot’s motion gait according to different environmental conditions (in this article, it focuses on tripod and trotting gait). Additionally, the reliability of the general method proposed in [30] was verified by experiments on quadruped and hexapod robots. From this, it can be seen that the two gaits use a six-link mechanism, which can significantly improve the motion speed of the hexapod robot. The tripod gait and bipod gait are not only applicable to hexapod robots but also to quadruped robots. However, we generally refer to “bipod gait” as trotting gait in quadrupedal robots, and bipod gait in hexapod robots. 

The main contributions of this paper are as follows:We proposed a six-link leg mechanism and applied it to a hexapod robot that can significantly improve the dynamic locomotion of the hexapod robot. We completed the kinematic modeling of this mechanism.We completed the single-legged foot-end trajectory planning of this hexapod robot based on the kinematic model. We introduced bipod gait into the gait planning of the hexapod robot, which enabled the leg mechanism to be more effective, and significantly improved the dynamic locomotion capability of the hexapod robot.We completed the motion simulation of a single leg and verified the high dynamic motion capability of this mechanism. We completed the physical prototype experiments to verify the feasibility of bipod gait with this mechanism to achieve highly dynamic walking of a hexapod robot.

This paper is organized as follows. Section 1 is an introduction. Section 2 presents an overview of the system. Section 3 presents an analysis of the robot motion, including the solution to the forward and inverse kinematics of the leg mechanism, single-leg trajectory planning, and gait planning. Section 4 presents the simulation and experiments of the robot. Section 5 is the conclusion. 

## 2. System Overview

### 2.1. Selection of Leg Mechanism

In this design, we adopted a six-link leg mechanism, which has the following advantages. First, the link structure has a significant bearing capacity, which can better support the robot’s body and keep the whole robot stable. Second, by simply changing the length of each element in the moving components, we can obtain different movement rules. Therefore, by designing the size and the whole structure of the connecting link, we can obtain leg movement rules similar to that of a robot. In addition, the light weight of the connecting rod also accelerates the movement speed of the robot. In a word, the linkage mechanism is suitable for the leg mechanism of a bionic robot. The movement diagram of the introduced leg mechanism is shown in Figure 1a.

This six-link mechanism can be regarded as a combination of two four-link mechanisms (1-2-3-4 and 1-4-5-6) with the same fixed frame. In Figure 1a, 1 is the fixed frame, and 2, the link, is the prime mover. Through the transmission of the connecting links 3, the rotation of the prime mover can be converted into the regular swing of link 4. The regular swing of link 4 can be converted into the lifting and lowering movement of link 6 by the transmission of the connecting rod 5. After calculation, the degree of freedom of the mechanism is 1, that is, the rotation angle of the prime mover and the foot end correspond with each other. This one-to-one correspondence is convenient for later debugging.

Figure 1b is a diagram of the leg mechanism. The leg mechanism includes a base numbered 2, a driving part numbered 1 arranged on the base, a connecting rod structure numbered 3 connected with the driving part, and a robot leg numbered 4 connected with the connecting rod structure.

Two bus-steering gears are fixed on the Z-shaped steering gear bracket to control the horizontal and vertical movements of the legs. The steering gear for controlling horizontal movement is connected with the upper and lower plates of the machine body through the steering gear plate, and the steering gear for controlling vertical movement is connected to the original parts of the leg linkage mechanisms through a group of parallelogram linkage mechanisms. Through the transmission of the leg six-bar linkage mechanism, the movement of the prime mover can be converted into the lifting movement of the foot end and then combined with the horizontal movement of the leg, thus achieving a leg-stepping action.

The main features of this mechanism are as follows:The specially designed linkage mechanism makes the leg move up and down within a wider range and the foot end easier to lift. In other words, when the steering gear rotates at the same angle, the foot end can move up a larger distance to enhance its ability to climb over obstacles in high-speed motion.The lighter weight of the linkage mechanism solves the problem of large joint mass.The centralized placement of steering gear lowers the leg’s moment of inertia and enhances its agility, providing the foundation for the high dynamic movement of the whole robot.This design is more convenient for wiring, which can better protect the steering gear to cope with more harsh environment.The U-shaped opening can reduce the mass and increase the working space, as the prime mover is frame-shaped, the connecting rod 34 is provided with a U-shaped opening facing the connecting rod 36, and the obtuse end of the connecting rod 36 is located in the U-shaped opening.Two degrees of freedom can reduce the difficulty of control.

Figure 2 is a diagram of the leg mechanism in two limit positions.

### 2.2. Selection of Body Base and Overall Mechanical Structure

In the above leg mechanism, the joint between the leg mechanism and the body base is relatively bloated due to the centralized placement of the steering gear, and the legs may interfere with each other during the movement. To give each leg more room to move so as not to interfere with each other, we adopted a design with an upper plate and a lower plate for the robot’s body base and sandwich the legs between the two plates. This not only solves the above problems but also makes the robot lighter and easier to disassemble.

The 3D model of the hexapod robot is shown in Figure 3.

## 3. The Theoretical Analysis of The Robot Movement

This section builds the kinematics analysis model of the robot leg, which lays the foundation for the trajectory planning of the foot end. We analyzed the planned foot trajectory and the foot trajectory to suit the characteristics of robot motion. Last, we built the dynamic model of the robot’s leg swing process to identify the real force in the robot’s movement process.

### 3.1. Kinematics Analysis

Figure 4 shows a local coordinate system with point *O* as the origin for analysis.

When the position of the prime mover is known, according to the geometric relationship of the mechanism, the coordinate position of the foot end *F* depends on the length of each rod and the position of the prime mover. The lengths of *OA*, *AB*, *BC*, *BD*, *CD*, *OE*, *OC*, *ED*, and *DF* are *l*_1_, *l*_2_, *l*_3_, *l*_4_, *l*_5_, *l*_6_, *s*, *a*, and *b*, respectively, and the angle between the prime mover and the positive direction of *Y-axis* is *θ*. To facilitate the solution, *AC* and *OC* are connected as auxiliary Δ AOC. Let $∠{OCA=\varphi }_{1},\;∠{ACB=\varphi }_{2},\;∠{COY=\alpha }_{1}$. Since *O* and *C* are fixed points, the coordinate expressions of *A* and *C* can be written directly.
(1)$$\left\{ \begin{array}{l} x_{A}=l_{1}\sin\theta \\ y_{A}=l_{1}\cos\theta \\ x_{C}=s\sin\alpha_{1}\\ y_{C}=s\cos\alpha_{1} \end{array}\right.$$

From the formula of the distance between two points in the plane coordinate system,
(2)$$l_{AC}=\sqrt{{\left(x_{A}-x_{C}\right)}^{2}+{\left(y_{A}-y_{C}\right)}^{2}}$$

∠OCA is always an acute angle when the prime mover swings in the range of motion. The value of *θ* is positive when *A* is on the right side of *Y-axis* and negative when *A* is on the left side of *Y-axis*. In ∆ *AOC*, it can be obtained by the sine theorem:(3)$$\varphi_{1}=\arcsin\frac{l_{1}\sin(\theta +\alpha_{1})}{l_{AC}}$$

In ∆ *ABC*, it can be obtained by the cosine theorem:(4)$$\varphi_{2}=\arccos\frac{l_{AC}^{2}+l_{3}^{2}-l_{2}^{2}}{2l_{AC}l_{3}}$$

$\theta_{1}{=\varphi }_{2}{\;-\;\varphi }_{1}$ can be obtained from Figure 4. In ∆ *OBC*, it can be obtained by the cosine theorem:(5)$$\left|OB\right|=\sqrt{s^{2}+l_{3}^{2}-2sl_{3}\cos\theta_{1}}$$

Points *O*, *B*, and *D* are collinear; therefore, we have the following:(6)$$\left|OD\right|=l_{4}+\sqrt{s^{2}+l_{3}^{2}-2sl_{3}\cos\theta_{1}}$$

After this, *OD* is referred to as $l_{8}$.

In ∆ *OCD*, $∠{OCD=\theta }_{1\;}{+\alpha }_{2},\;∠{ODC=\alpha }_{3}$,$∠COD=\pi \;-{\;\theta }_{1}\;-\;\alpha_{2}-\alpha_{3}$; therefore, we obtain the following:(7)$$∠DOX=∠COD-∠COX=\pi -\theta_{1}-\alpha_{2}-\alpha_{3}-(\frac{\pi }{2}-\alpha_{1})=\frac{\pi }{2}-\theta_{1}+\alpha_{1}-\alpha_{2}-\alpha_{3}$$

After this, ∠DOX is referred to as $\theta_{2}$.

The coordinates of point *D* can be obtained from Figure 4.
(8)$$\left\{ \begin{array}{l} x_{D}=l_{8}\cos\theta_{2}\\ y_{D}=l_{8}\sin\theta_{2} \end{array}\right.$$

For point *E*, it moves in a circle around point *O*, and the distance between it and point *D* is *a*, so we can obtain the equations about the coordinates of point *E*.
(9)$$\left\{ \begin{array}{l} x_{E}^{2}+y_{E}^{2}=l_{6}^{2}\\ {\left(x_{E}-x_{D}\right)}^{2}+{\left(y_{E}-y_{D}\right)}^{2}=a^{2} \end{array}\right.$$

Since *E*, *D*, and *F* are collinear, assuming *a* = *λb*, the coordinate expression of the foot-end *F* point can be obtained by the formula of fixed ratio point coordinates.
(10)$$\left\{ \begin{array}{l} x_{F}=\frac{(1+\lambda )x_{D}-x_{E}}{\lambda }\\ y_{F}=\frac{(1+\lambda )y_{D}-y_{E}}{\lambda } \end{array}\right.$$

As shown in Figure 4, *λ* = *a*/*b*, so the coordinate expression of the foot end *F* point is as follows:(11)$$\left\{ \begin{array}{l} x_{F}=\frac{(a+b)x_{D}-bx_{E}}{a}\\ y_{F}=\frac{(a+b)y_{D}-by_{E}}{a} \end{array}\right.$$

### 3.2. Foot Trajectory Planning

Here, the requirement of the foot end trajectory is as follows.

The trajectory of the foot should be a continuous and smooth closed curve, with no sudden change in speed and acceleration.There is no impact between the leg movement and the ground, that is, the speed and acceleration when landing and leaving the ground are 0.Try to avoid unnecessary exercise.

The legs in the supporting phase bear the body’s weight and the robot’s extra load, generating body movement through relative movement. The trajectory projection formed by the leg in the supporting phase contacting the ground is mostly a simple straight line. Swing is the continuous movement process of feet from the ground up to the ground down, which determines the step length and height of the whole robot and has an essential influence on the robot’s motion performance. Typical robot foot trajectory curves include rectangular curve, elliptic curve, parabola, modified cycloid, heart-shaped line, combined line segment, etc. [32,33,34]. The functional forms of cycloid and elliptic curves are elementary, and their derivatives are continuous and smooth and have no mutation [35]. The starting angle and landing angle of the swing phase are right angles, there is no horizontal velocity component, and it is not easy to slip. Therefore, there is no sudden change in speed during walking, which can ensure that the robot walks stably without impact. The expressions for correcting cycloid and elliptic curves are as follows.

Modified cycloid expression:(12)$$\left\{ \begin{array}{l} x=\frac{L}{2\pi }(\varphi -\sin\varphi )\\ y=\frac{H}{2}(1-\cos\varphi ) \end{array}\right.$$

Elliptic curve expression:(13)$$\left\{ \begin{array}{l} x=\frac{L}{2}\cos\varphi +\frac{L}{2}\\ y=\frac{H}{2}\sin\varphi +\frac{H}{2} \end{array}\right.$$
where *L*, *H*, and *φ* are step length, step height, and polar angle, respectively.

Figure 5 shows the trajectory comparison between the elliptic curve and modified cycloid, in which the step length *L* = 100 mm and the step height *H* = 40 mm. As can be seen from the figure, both curves are continuous and smooth, with no abrupt change. Furthermore, with the same step length and step height, the trajectory of the modified cycloid is shorter than that of the elliptic curve, which reduces unnecessary motion and consumes less energy. Therefore, this paper uses the modified cycloid to plan the foot-end trajectory.

It can also be found that, in the whole trajectory period, the modified cycloid and ellipse have sharp points at the junction with the straight line (point *A* and point *B*), which means that there is a sudden change at the transition between the swing phase and the support phase, which will significantly affect the stability of robot motion.

It is necessary to make a smooth transition between the supporting and swinging phases to make the foot end track continuous and smooth. In this paper, the method of quintic polynomial interpolation is used to smooth the transition of the foot-end trajectory composed of the modified cycloid and straight lines. Because of the symmetry of the trajectory, the proper transition function can be written by the left transition function. The specific planning method is as follows.

The length *R* is subtracted from the left end of the swing phase and the left end of the support phase to construct the retraction transition curve. As shown in Figure 6, the trajectory consists of $\hat{BC},\;\hat{CD},\;\hat{DE},\;\hat{EB}$.Let the coordinates of point *B* and point *C* be $B\left(\varphi_{0}{,\;x}_{0}{,\;y}_{0}\right)$ and $C\left(\varphi_{1}{,\;x}_{1}{,\;y}_{1}\right)$, and their velocities and accelerations can be expressed as $\dot{B}\left(\varphi_{0},\;\dot{x_{0}},\;\dot{y_{0}}\right)$, $\ddot{B}\left(\varphi_{0},\;\ddot{x_{0}},\;\ddot{y_{0}}\right)$, $\dot{C}\left(\varphi_{1},\;\dot{x_{1}},\;\dot{y_{1}}\right)$ and $\ddot{C}\left(\varphi_{1},\;\ddot{x_{1}},\;\ddot{y_{1}}\right)$. Then, from the coordinates of points *B* and *C*, the expression of the trajectory can be written as a quintic polynomial function.

From *B* to *C*:(14)$$x(\varphi )=\left[\begin{array}{cccc} a_{1} & a_{2} & a_{3} & a_{4}\;\begin{array}{cc} a_{5} & a_{6} \end{array} \end{array}\right]{\left[\begin{array}{cccc} \varphi^{5} & \varphi^{4} & \varphi^{3} & \varphi^{2}\;\begin{array}{cc} \varphi & 1 \end{array} \end{array}\right]}^{T}$$
(15)$$y(\varphi )=\left[\begin{array}{cccc} b_{1} & b_{2} & b_{3} & b_{4}\;\begin{array}{cc} b_{5} & b_{6} \end{array} \end{array}\right]{\left[\begin{array}{cccc} \varphi^{5} & \varphi^{4} & \varphi^{3} & \varphi^{2}\;\begin{array}{cc} \varphi & 1 \end{array} \end{array}\right]}^{T}$$

The value range of φ in Equations (14) and (15) is 0 ≤ φ ≤ Φ. 

From *C* to *D*:(16)$$\left\{ \begin{array}{l} x(\varphi )=\frac{L}{2\pi }(2\varphi -\sin2\varphi )\\ y(\varphi )=\frac{H}{2}(1-\cos2\varphi ) \end{array}\right.(\Phi \leq \varphi \leq \pi -\Phi )$$

From *D* to *E*:(17)$$x(\varphi )=L-\left[a_{1}\cdots a_{6}\right]{\left[\begin{array}{cc} {\left(\pi -\varphi \right)}^{5} & {\left(\pi -\varphi \right)}^{4}\cdots \begin{array}{c} 1 \end{array} \end{array}\right]}^{T}\;(\pi -\Phi \leq \varphi \leq \pi )$$
(18)$$y(\varphi )=\left[b_{1}\cdots b_{6}\right]{\left[\begin{array}{cc} {\left(\pi -\varphi \right)}^{5} & {\left(\pi -\varphi \right)}^{4}\cdots 1 \end{array}\right]}^{T}\;(\pi -\Phi \leq \varphi \leq \pi )$$

The expressions of velocity and acceleration can be obtained by taking the first derivative and the second derivative of the expression of *BC*, respectively:


(19)
$$\dot{x}(\varphi )=\left[\begin{array}{ccccc} 5a_{1} & 4a_{2} & 3a_{3} & 2a_{4} & a_{5} \end{array}\right]{\left[\begin{array}{ccccc} \varphi^{4} & \varphi^{3} & \varphi^{2} & \varphi & 1 \end{array}\right]}^{T}\;(0\leq \varphi \leq \Phi )$$



(20)
$$\dot{y}(\varphi )=\left[\begin{array}{ccccc} 5b_{1} & 4b_{2} & 3b_{3} & 2b_{4} & b_{5} \end{array}\right]{\left[\begin{array}{ccccc} \varphi^{4} & \varphi^{3} & \varphi^{2} & \varphi & 1 \end{array}\right]}^{T}\;(0\leq \varphi \leq \Phi )$$



(21)
$$\ddot{x}(\varphi )=\left[\begin{array}{cccc} 20a_{1} & 12a_{2} & 6a_{3} & 2a_{4} \end{array}\right]{\left[\begin{array}{cccc} \varphi^{3} & \varphi^{2} & \varphi & 1 \end{array}\right]}^{T}\;(0\leq \varphi \leq \Phi )$$



(22)
$$\ddot{y}(\varphi )=\left[\begin{array}{cccc} 20b_{1} & 12b_{2} & 6b_{3} & 2b_{4} \end{array}\right]{\left[\begin{array}{cccc} \varphi^{3} & \varphi^{2} & \varphi & 1 \end{array}\right]}^{T}\;(0\leq \varphi \leq \Phi )$$


Similarly, the expressions of velocity and acceleration can be obtained by taking the first derivative and the second derivative of the expression of *CD* segment, respectively:(23)$$\left\{ \begin{array}{l} \dot{x}(\varphi )=\frac{L}{\pi }(1-\cos2\varphi )\\ \dot{y}(\varphi )=H\sin2\varphi \\ \ddot{x}(\varphi )=\frac{2L}{\pi }\sin2\varphi \\ \ddot{y}(\varphi )=2H\cos2\varphi \end{array}\right.(\Phi \leq \varphi \leq \pi -\Phi )$$

The phase of point *C* is *π/9*, and the coordinates of point *B* are (0, 25/9, 0), from which the coordinates of $\dot{B}$, $\ddot{B}$, $\dot{C}$ and $\ddot{C}$ can be written. Substituting the coordinates of *B*, *C*, $\dot{B}$, $\ddot{B}$, $\dot{C}$ and $\ddot{C}$ into (14), (15), and (19)–(22), the coefficients of the quintic polynomial can be obtained simultaneously as follows.
(24)$$\left[\begin{array}{ccc} a_{1} & a_{2} & a_{3}\\ a_{4} & a_{5} & a_{6} \end{array}\right]=\left[\begin{array}{ccc} -3220.72 & 2806.31 & -627.61\\ 0 & 0 & 2.78 \end{array}\right]$$
(25)$$\left[\begin{array}{ccc} b_{1} & b_{2} & b_{3}\\ b_{4} & b_{5} & b_{6} \end{array}\right]=\left[\begin{array}{ccc} 6359.48 & -5726.21 & 1443.96\\ 0 & 0 & 0 \end{array}\right]$$

It should be noted that the foot trajectory planning method [36] introduced in this paper is aimed at motion in a two-dimensional plane. Because of the characteristics of the linkage mechanism, the foot of the robot introduced in this paper moves in three-dimensional space, so the foot trajectory planning described in this paper is the foot trajectory projection planning. In addition, since the installation position of the robot leg on the body base is determined in this paper, if the robot needs to walk in a straight line when moving forward or backward, the step length of each group of legs will be different, but as long as the method introduced in this paper is followed, the changed trajectory parameters can be obtained.

### 3.3. Leg Dynamics Analysis

In this paper, the ground reference system is used to establish the coordinate system, as shown in Figure 7. Take the generalized joint variable $q=\;{\left[\theta_{1}{,\;\theta }_{2}{,\;\theta }_{3}{,\;\theta }_{4}{,\;\theta }_{5}{,\;\theta }_{6}{,\theta }_{7}{,\;\theta }_{8}\right]}^{T}$. The center point of each rod is located in the geometric center of the rod. The corresponding generalized driving force *F* can be obtained by using the Lagrange equation:(26)$$F=\frac{d}{dt}(\frac{\partial E_{K}}{\partial {\dot{q}}_{i}})-\frac{\partial E_{K}}{\partial q_{i}}+\frac{\partial E_{P}}{\partial q_{i}}$$
where $E_{K}$ is the total kinetic energy of the system, $E_{P}$ is the total potential energy of the system, and $q_{i}$ is the generalized coordinate.

The leg vector of the robot is $P_{H}$, the acceleration vector of gravity is ***g***, the center vector of each link is $P_{i}$, the mass of each component is $m_{i}$, the velocity vector of the center of mass is $v_{i}$, the angular velocity vector is $\omega_{i}$, and the inertia tensor is $I_{i}$. Then, the total kinetic energy of the whole system is the sum of the kinetic energy of each connecting rod. After homogenizing each connecting rod, calculating the centroid coordinates, and taking the derivative, the following formula can be obtained:(27)$$E_{K}=\frac{1}{2}(m_{i}v_{i}^{T}v_{i}+\omega_{i}^{T}I_{i}\omega_{i})$$

Through simultaneous (26) and (27), the joint torque can be obtained:(28)$$\tau =D(q)\ddot{q}+H(q,\dot{q})+G(q)$$
where D(q) is the inertial matrix of the robot legs, $H\left(q,\ddot{q}\right)$ is the Coriolis force vector, and G(q) is the gravity vector.

If the gait cycle of the robot is *T*, the energy consumed by the robot in one gait cycle can be expressed as follows:(29)$$E=\int_{0}^{T}\tau \dot{q}dt$$

### 3.4. Gait Planning and Analysis

For the hexapod robot described in this paper, the movement gait is a typical crawling principle of hexapod insects. In order to clearly describe the movement law of legs, the schematic diagram and timing diagram are now used to describe the step sequence of the hexapod robot. As shown in Figure 8, the six legs of a hexapod robot are divided into two groups: defined and numbered. The left front foot, right middle foot, and left rear foot (*L1*, *R2*, and *L3*) of the hexapod robot are a group, and the right front foot, left middle foot, and right rear foot (*R1*, *L2*, and *R3*) are a group. The three legs of each group can form a stable triangular structure, so it is called a tripod gait.

The basic movements of the hexapod robot can be achieved by the different alternations of the two groups of legs. When the hexapod robot moves forward, *R1*, *R3*, and *L2* move towards the head and are in the supporting phase, while *L1*, *L3*, and *R2* move towards the tail and are in the swinging phase. Then, the two groups of legs alternately take the above actions to complete the hexapod robot’s advance. The timing chart of the advance is shown in Figure 9.

Similarly, when the hexapod robot retreats, it only needs to change the rotation direction and to run the logic for each leg. For a hexapod robot, the rotation direction of each group of legs is slightly different when it turns left and right on a fixed axis. For example, when it turns left on a fixed axis, *R1*, *L2*, and *R3* rotate counterclockwise in the supporting phase and *L1*, *R2*, and *L3* rotate clockwise in the swinging phase, and then the two groups of legs alternately take the above actions. The same is true for turning right on a fixed axis.

In nature, multi-legged insects have a typical tripod gait, but studies have shown that, when there is no adhesive structure on the legs of insects, the bipod gait similar to vertebrates performs better than the tripod gait in terms of movement speed [37]. Bipedal gait refers to dividing the legs of a hexapod robot into three groups (the first group is *L1* and *R3*, the second group is *L2* and *R2*, and the third group is *R1* and *L3*) and ensuring that only the legs of the same group land simultaneously every time the hexapod robot lands. As only two legs of the hexapod robot land on the ground at the same time, compared with the tripod gait, the bipod gait has no static stability but is a dynamically balanced gait. With a high frequency of alternating movements of each group of legs, the bipod gait equips the hexapod robot with high dynamic movement capability.

When the robot moves in a bipod gait, *L2* and *R2* (group 2), and *R1* and *L3* (group 3) move towards the head of the robot in a swinging phase, and *L1* and *R3* (group 1) move towards the tail in a supporting phase; then, each group of legs moves in turn according to this logic. The timing chart of advance is shown in Figure 10.

### 3.5. Analysis of Gait Stability of Hexapod Robot

This section provides an analysis of the tripod gait as the static gait and an analysis of the bipod gait as the dynamic gait.

In this paper, the projection of the center of gravity on the *XOY* plane is used to determine whether the robot meets the static conditions. In the tripod gait, at most three legs are in the supporting phase. As shown in Figure 11, when the robot moves forward, it is assumed that *L1*, *L3*, and *R2* are in the supporting phase and that *R1*, *R3*, and *L2* are in the swinging phase at a certain moment. At this time, the endpoints *S1*, *S2*, and *S3* of the three legs in the supporting phase will form a triangle. When the projection point *P* of the robot’s center of gravity *M* on *XOY* plane is located in the triangle area, the static gait stability condition is satisfied.

The mathematical relationship after converting the stability conditions corresponding to the robot stability is ${△}_{S1S2S3}$, and the sum of the areas of the three triangles divided by point *P* is equal to ${△}_{S1S2S3}$. The specific geometric analysis is shown in Figure 12.

The coordinates of *S*1, *S*2, *S*3, and point *P* of the robot can be known by moving in the *XOY* plane, and the coordinates are set as *S*1($x_{1}$,$y_{1}$), *S*2($x_{2}$,$y_{2}$), *S*3($x_{3}$,$y_{3}$), and *P* ($x_{P}$,$y_{P}$
), respectively, so that the areas of the four triangles can be obtained.


(30)
$$\left[\begin{array}{cc} S_{△S1S2P} & S_{△S1S3P}\\ S_{△S2S3P} & S_{△S1S2S3} \end{array}\right]=\left[\begin{array}{cc} \left|\frac{1}{2}\times \left|\begin{array}{ccc} x_{1} & y_{1} & 1\\ x_{2} & y_{2} & 1\\ x_{p} & y_{p} & 1 \end{array}\right|\right| & \left|\frac{1}{2}\times \left|\begin{array}{ccc} x_{1} & y_{1} & 1\\ x_{3} & y_{3} & 1\\ x_{p} & y_{p} & 1 \end{array}\right|\right|\\ \left|\frac{1}{2}\times \left|\begin{array}{ccc} x_{2} & y_{2} & 1\\ x_{3} & y_{3} & 1\\ x_{p} & y_{p} & 1 \end{array}\right|\right| & \left|\frac{1}{2}\times \left|\begin{array}{ccc} x_{1} & y_{1} & 1\\ x_{2} & y_{2} & 1\\ x_{3} & y_{3} & 1 \end{array}\right|\right| \end{array}\right]$$


The mathematical expression for keeping the robot’s static gait stable is as follows:(31)$$S_{△S1S2S3}=S_{△S1S2P}+S_{△S1S3P}+S_{△S2S3P}$$

If (31) is satisfied, the center of gravity of the robot falls in the area of ${△}_{S1S2S3}$ and the stability condition is satisfied. In the tripod gait, the full-tube support triangle will change with the cycle, but the projection point *P* of the robot’s center of gravity on the *XOY* plane will always be located in the support triangle, so the robot in the tripod gait is always in a stable state.

In order to make the hexapod robot perform better in high-speed movement, this paper adopts a dynamic gait that is bipodal. However, the bipod gait is more complicated than the tripod gait, and the stability analysis is relatively complicated. The ZMP theory is used to analyze the stability of robot bipod dynamic gait [38].

In dynamic motion, the *R1* and *L3* legs are in the supporting phase and the other four legs are in the swinging phase; the dynamic motion stability of the robot can be analyzed. As shown in Figure 13, the coordinate relationship of ZMP can be obtained from Figure 13a.
(32)$$\left\{ \begin{array}{l} \frac{F_{Z}}{F_{X}}=\frac{z_{A}}{x_{A}-x_{ZMP}}\\ \frac{F_{Z}}{F_{Y}}=\frac{z_{A}}{y_{A}-y_{ZMP}} \end{array}\right.$$

In Formula (32), $F_{X}$, $F_{Y}$, and $F_{Z}$ are the components of the resultant force of the robot in *X*, *Y*, and *Z* directions, respectively. $x_{A}$, $y_{A}$, and $z_{A}$ are the respective coordinates of the center of gravity of the robot in *X*, *Y*, and *Z* directions. $x_{ZMP}$ and $y_{ZMP}$ are the respective coordinates of ZMP in the *X* and *Y* directions.

According to mechanics, the resultant force of the robot in motion is the resultant force of inertia force and gravity, so the following expression can be obtained:(33)$$F=\left[\begin{array}{c} F_{X}\\ F_{Y}\\ F_{Z} \end{array}\right]=\sum_{i=1}^{n}m_{i}\left[\begin{array}{c} {\ddot{x}}_{i}\\ {\ddot{y}}_{i}\\ {\ddot{z}}_{i}+g \end{array}\right]$$

In Formula (33), $m_{i}$ is the weight of each component; $\ddot{x_{i}}$, $\ddot{y_{i}}$ and $\ddot{z_{i}}$ are the acceleration rates of each component in the *X*, *Y*, and *Z* directions. The acceleration of gravity is *g*.

Simultaneously using Equations (32) and (33), the coordinates of ZMP point can be obtained:(34)$$\left\{ \begin{array}{l} x_{ZMP}=\frac{\sum_{i=1}^{n}m_{i}({\ddot{z}}_{i}+g)x_{A}-\sum_{i=1}^{n}m_{i}{\ddot{x}}_{i}z_{A}}{\sum_{i=1}^{n}m_{i}({\ddot{z}}_{i}+g)}\\ y_{ZMP}=\frac{\sum_{i=1}^{n}m_{i}({\ddot{z}}_{i}+g)y_{A}-\sum_{i=1}^{n}m_{i}{\ddot{y}}_{i}z_{A}}{\sum_{i=1}^{n}m_{i}({\ddot{z}}_{i}+g)} \end{array}\right.$$

In order to make the robot move dynamically without tipping over, ZMP should fall on the line connecting the foot ends of the two supporting legs. In this case, the stable condition of the robot’s dynamic motion is that ZMP should fall on the connecting line between *R*1 and *L*3, and Equation (35) should be satisfied:(35)$$(y_{3}-y_{4})x_{ZMP}+(x_{4}-x_{3})y_{ZMP}+(x_{3}y_{4}-x_{4}y_{3})=0$$

When the robot moves with a bipod gait, it is difficult for the center of gravity of the robot to fall onto the connecting line between the two supporting legs, so the method of judging the stability condition by zero moment point will fail. It takes a certain time for the robot to move from steady-state to dumping. If the robot can adjust its posture during this time, it can avoid dumping. Therefore, when the movement gait is determined, the subsequent posture sequence of each posture of the robot can be estimated, so as to calculate the time *T_1_* required for the robot to move from a stable state to an unbalanced state. If the response time *T* of the robot to adjust its posture is shorter than *T_1_*, the robot can complete the stable adjustment and realize the dynamic stability.

## 4. Simulation and Experimental Analysis

### 4.1. One-Leg Simulation Analysis

In this section, the SolidWorks Motion plug-in was used to simulate and analyze the motion characteristics of the robot. The input analog were data points of the servo turning angle calculated from the inverse kinematics, with the interpolation type Akima spline curve. The single-leg simulation results are presented in Figure 14. The maximum lifting speed of the foot end could reach 86 mm/s (Figure 14b), while the maximum lifting acceleration of the foot end could reach 413 mm/s^2^ (Figure 14c). This result verified the high dynamic motion capability of this mechanism. Figure 14b, c also show that there was no sudden change in both velocity and acceleration of the foot end during the motion. This result verified the correctness of the foot-end trajectory planning method described in Section 3. Table 1 shows the material information table for each component. Figure 15 shows the trajectory of the foot end in this simulation.

### 4.2. Physical Prototype Experiments

In the field experiment, the robot was equipped with a nine-axis IMU for acceleration measurement, and the rotation speed of the servo was tuned to 0.2 r/s (the maximum speed of servo operation). The single-leg trajectory planning scheme presented in Section 3 was introduced into the robot, and the robot moved forward for 20 s on flat ground in the laboratory with the tripod gait and the bipod gait. Figure 16 is the experimental environment of the robot.

As shown in Figure 11, the forward direction of the robot was the positive direction of the *X-axis*. First, the robot moved forward with a tripod gait. Figure 17a is an image of the acceleration of the robot moving along the *X-axis* changing with time. The acceleration of the robot showed a periodic trend of first increasing, then decreasing, and then increasing. The reason for this is that the robot’s action switching in the tripod gait involved the acceleration and deceleration processes, and thus, the ideal uniform linear motion did not appear. The absolute value of the acceleration of the robot in the *X-axis* direction could reach 4.1 mm/s^2^, which indicates that the robot had high dynamic motion capability.

Figure 17b is an image of the acceleration of the robot moving along the *X-axis* with bipod gait. When the robot moved forward with a bipod gait, the frequency of change in acceleration and speed was higher, and the absolute peak value of acceleration reached 7.8 mm/s^2^, which put the robot in a state of dynamic equilibrium and improved its dynamic motion capability.

To further observe the motion performance of the robot moving forward with two gaits, we integrated the acceleration images of Figure 17a, b once to derive the velocity images of both, and the results are shown in Figure 18. Comparing Figure 18a,b, it could be seen that the robot moves forward with a tripod gait with lower frequency and lower amplitude, and the velocity varies slightly around the average value, which also proves the superior stability of the tripod gait.

The advantage of the bipod gait is the dynamic stability and higher speed: On the one hand, the high frequency of speed changes reflects the dynamic stability of the robot. On the other hand, from Figure 18b, it could be found that the robot reached a peak velocity of 390 mm/s (130% of the moveable body length per second) and an average velocity of 174 mm/s (58% of the moveable body length per second) during 20 s of movement, which is 49.5% higher than the peak velocity and 34.5% higher than the average velocity of the tripod gait. However, the robot’s motion speed fluctuated considerably above and below the average value, so we could see that the robot’s speed improved when it moved with a bipod gait, but its motion stability was not satisfactory. Another set of data should be used to measure the stability of the robot’s motion when walking with different gaits.

Ideally, the motion of the robot on the horizontal plane should be in a horizontal stable state; that is, the displacement in the *Z-axis* direction is zero, but this was not the case in practice. The acceleration in the *Z-axis* direction can be integrated twice to obtain the displacement of the robot on the *Z-axis*, which represents the motion turbulence of the robot along the *Z-axis*. First, the robot moved forward with a tripod gait. The displacement of the robot on the *Z-axis* in the actual movement process is shown in Figure 19a. The absolute value of the maximum forward displacement of the robot on the *Z-axis* is 3.5 mm (equivalent to 2.5% of body height), the absolute value of maximum reverse displacement is 8.5 mm (equivalent to 6.1% of body height), and the absolute value of average value is 3.7 mm (equivalent to 6.1% of body height). The reason for this situation is that the movement conversion of the robot in the forward movement with a tripod gait involved the lifting and lowering movement conversion of the legs, which led to the fact that the robot’s center of gravity was not always at the same horizontal height. Consequently, the robot bumped up and down slightly along the *Z-axis*. Finally, the degree of motion turbulence in the *Z-axis* direction of the robot is reflected by variance:(36)$$\sigma_{1}^{2}=\frac{\sum {\left(Z-{\bar{Z}}_{1}\right)}^{2}}{n-1}=12.21$$

Figure 19b is the displacement change on the *Z-axis* when the robot advanced with the bipod gait. From the image, it can be seen that the maximum displacement and the average displacement of the robot on the *Z-axis* were both larger. This is because the robot’s motion changed frequently in the bipod gait, which enabled it to achieve a dynamic balance. Its variance is as follows:(37)$$\sigma_{2}^{2}=\frac{\sum {\left(Z-{\bar{Z}}_{2}\right)}^{2}}{n-1}=47.61$$

By comparison, it can be seen that the robot’s degree of bumpiness on the *Z-axis* was far lower and that its performance was more stable when walking in the tripod gait than in the bipod gait. However, regardless of the bipod gait or tripod gait being used, the robot did not show excellent stable movement, and there was also a big fluctuation in the stable tripod gait. This is because the parts of the leg mechanism were mostly 3D-printed, and the overall stiffness of the leg was low. Consequently, there was a big vertical and horizontal vibration in the robot movement.

## 5. Conclusions

In this paper, we introduced a design method and implemented a hexapod robot; the contribution of this paper are as follows. First, a six-link mechanism was designed. This mechanism can realize the transformation from small driving motion to large output motion, thus enhancing the dynamic motion capability of a legged robot. Second, the robot single-leg trajectory planning and a planning method for tripod and bipod gait are proposed and verified for feasibility. The tripod gait ensures the stability of the robot running on an unstructured terrain, while the bipod gait ensures the high speed of the robot on a flat terrain. Simulations and experiments verified that our method can improve the dynamic motion capability of the hexapod robot.

Future work will enhance the hexapod robots’ sensing ability and establish a dynamic gait transition method for hexapod robots on various unstructured terrains. 

## Figures and Tables

**Figure 1 micromachines-13-01404-f001:**
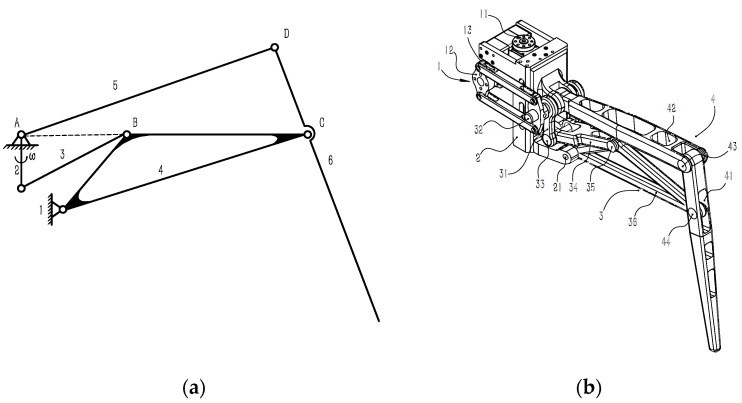
(**a**) The skeleton of the leg mechanism. (**b**) The leg mechanism diagram.

**Figure 2 micromachines-13-01404-f002:**
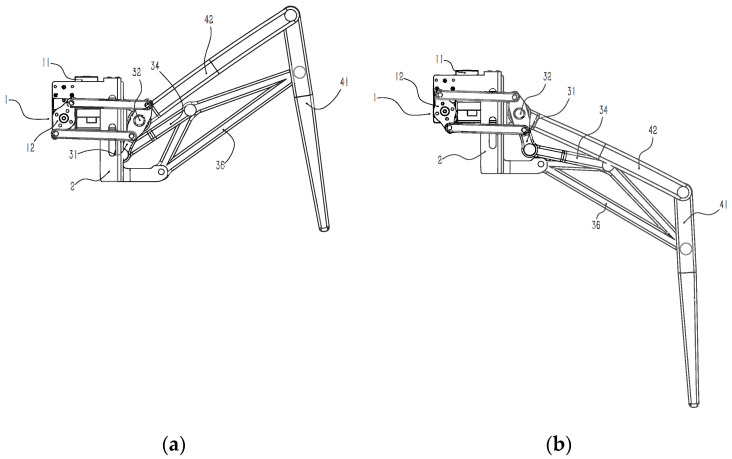
(**a**) A diagram of the leg mechanism moving to the lowest position and (**b**) a diagram of the leg mechanism moving to the highest position.

**Figure 3 micromachines-13-01404-f003:**
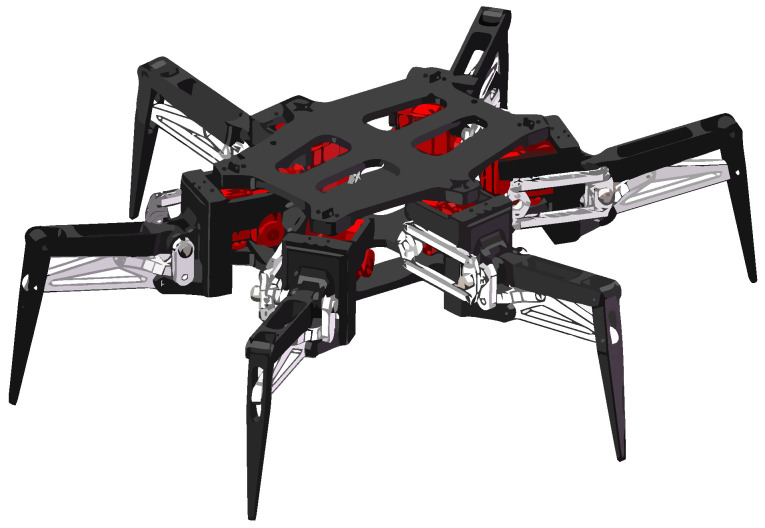
Three-dimenasional model of the robot.

**Figure 4 micromachines-13-01404-f004:**
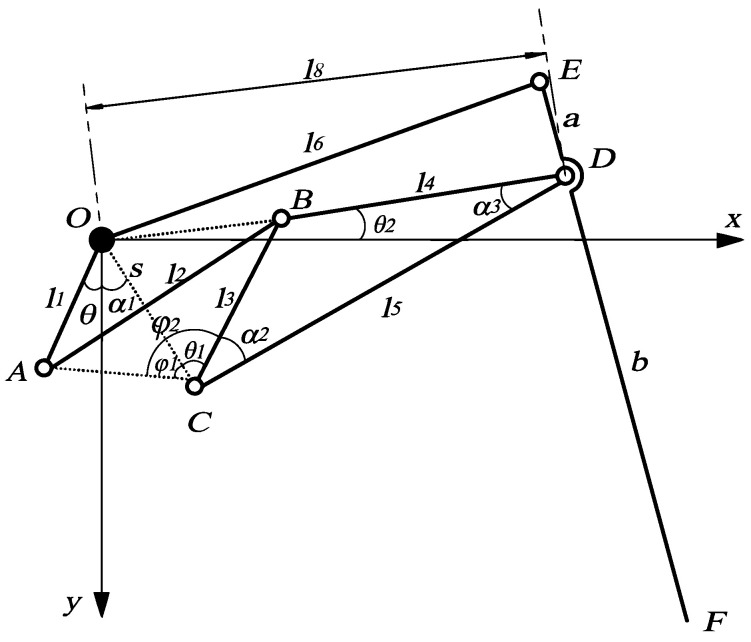
Schematic diagram of kinematics analysis.

**Figure 5 micromachines-13-01404-f005:**
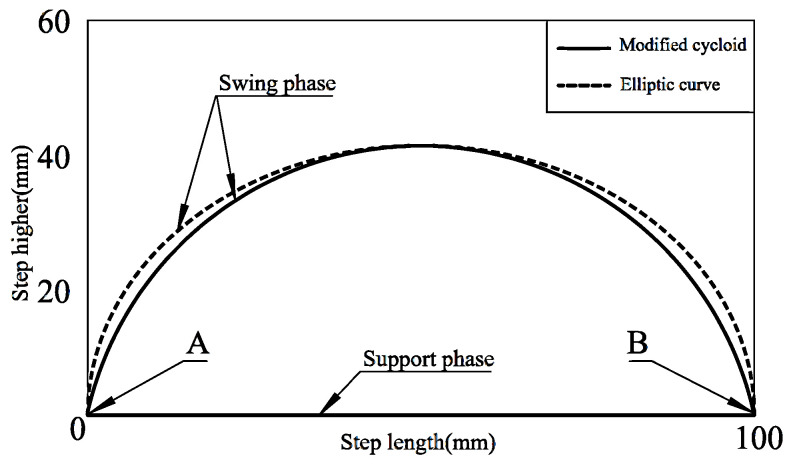
Correction of cycloid and elliptic curves.

**Figure 6 micromachines-13-01404-f006:**
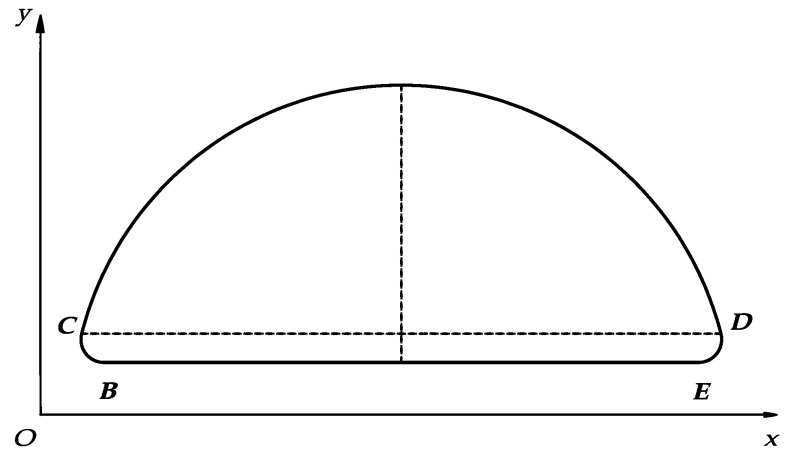
Optimized foot trajectory.

**Figure 7 micromachines-13-01404-f007:**
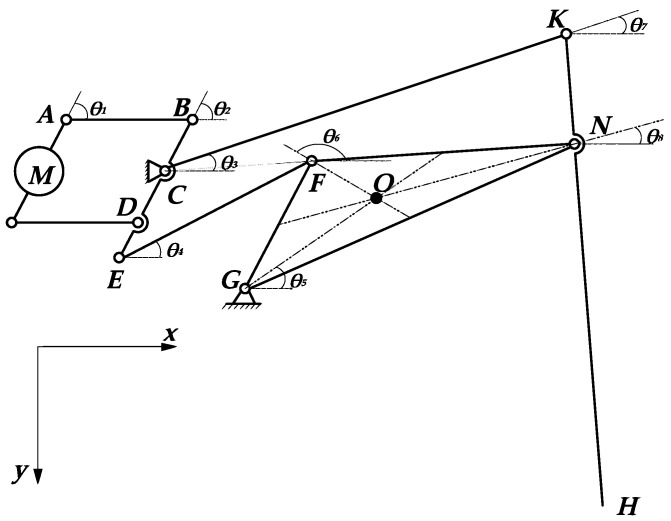
Leg dynamics analysis model.

**Figure 8 micromachines-13-01404-f008:**
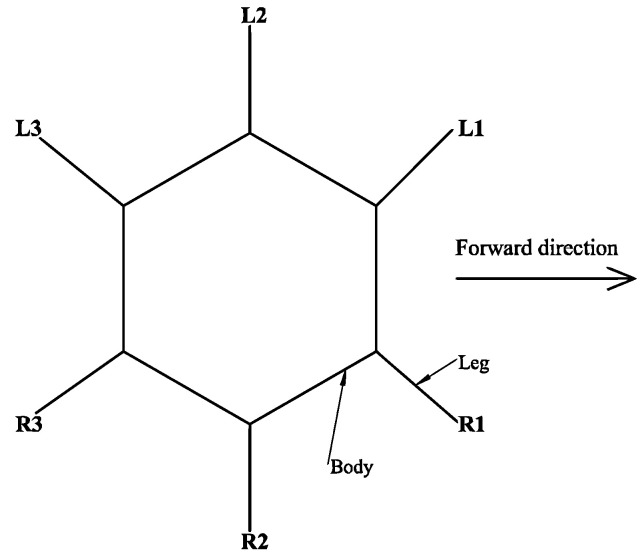
Gait analysis model.

**Figure 9 micromachines-13-01404-f009:**
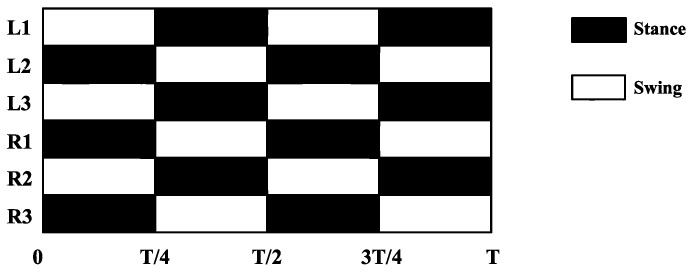
Sequence diagram of tripod gait.

**Figure 10 micromachines-13-01404-f010:**
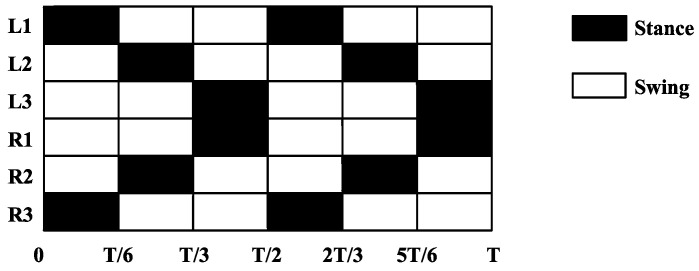
Sequence diagram of bipod gait.

**Figure 11 micromachines-13-01404-f011:**
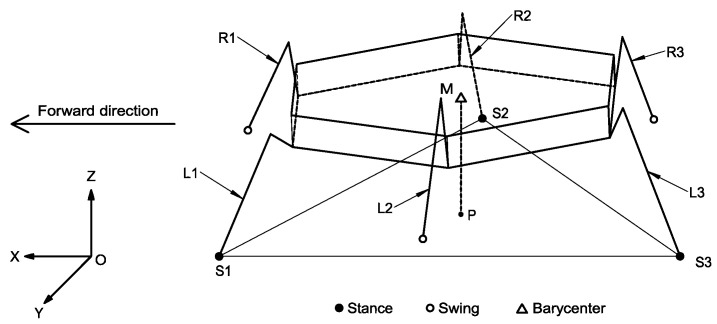
Geometric analysis of static gait stability.

**Figure 12 micromachines-13-01404-f012:**
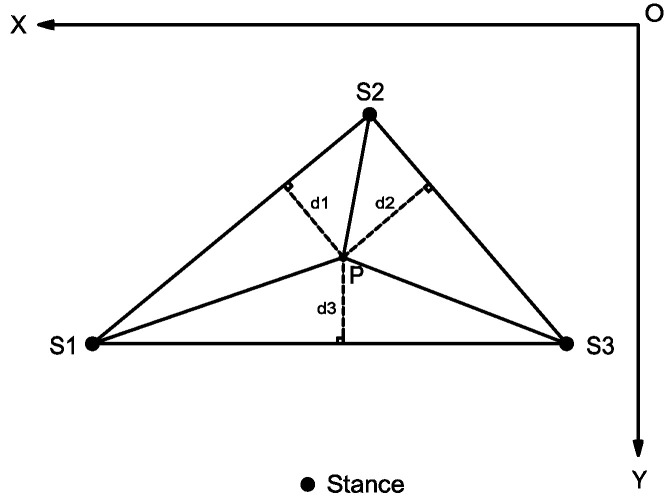
Analysis of mathematical relationship for static gait stability.

**Figure 13 micromachines-13-01404-f013:**
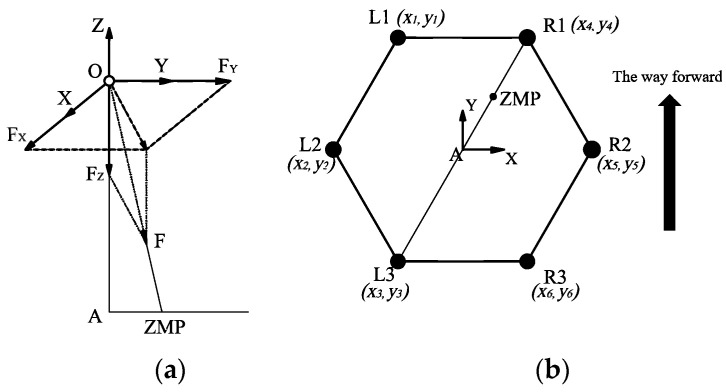
The stability analysis of robot bipod gait. (**a**) The schematic diagram of ZMP point location. (**b**) The stability analysis model of bipod gait.

**Figure 14 micromachines-13-01404-f014:**
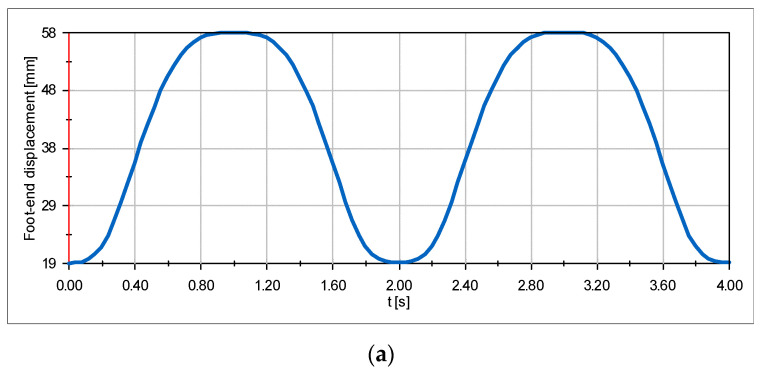
The result of the simulation experiment for a single leg. (**a**) The displacement graph of the foot end on the *Y-axis*. (**b**) The velocity graph of the foot end on the *Y-axis*. (**c**) The acceleration graph of the foot end on the *Y-axis*.

**Figure 15 micromachines-13-01404-f015:**
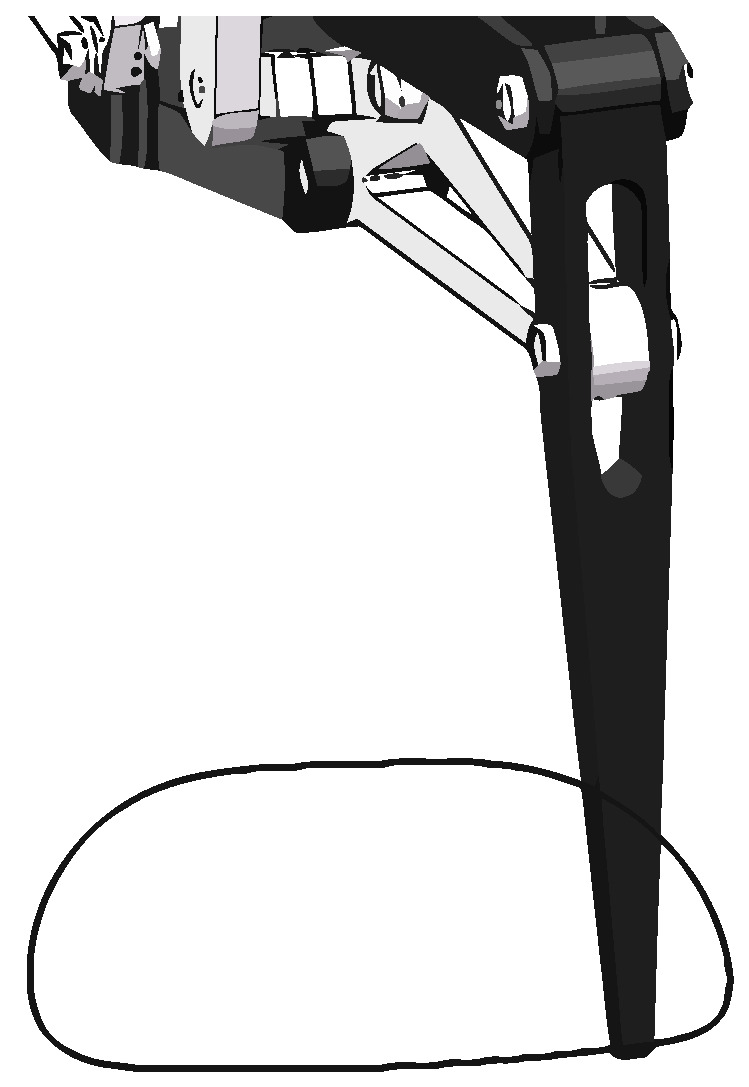
Foot simulation trajectory.

**Figure 16 micromachines-13-01404-f016:**
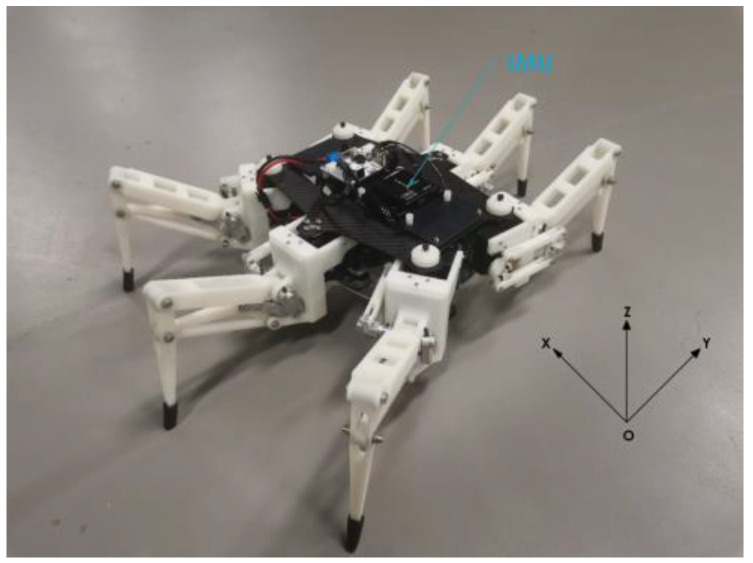
Robot experiment scene.

**Figure 17 micromachines-13-01404-f017:**
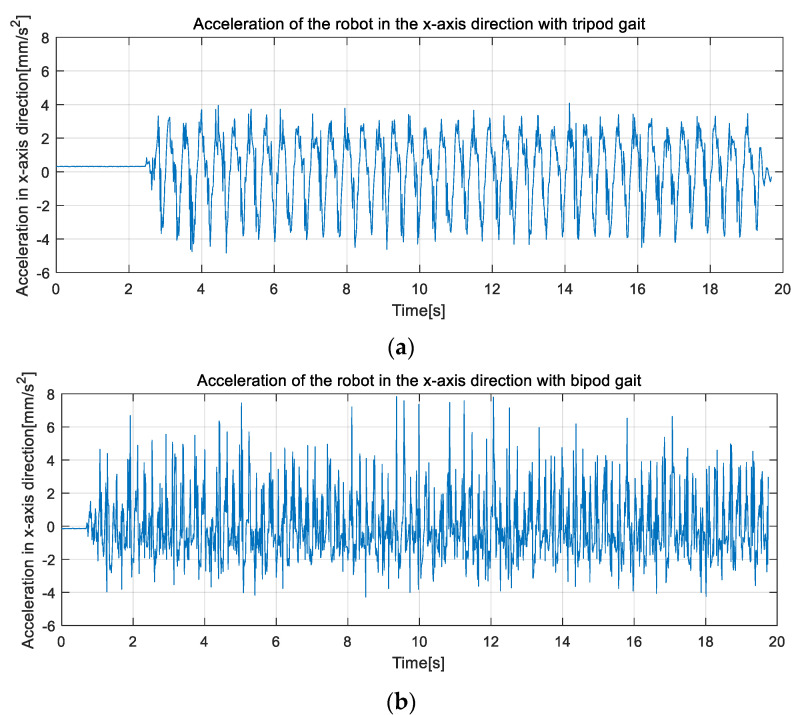
The experimental result of the physical prototype. (**a**) The acceleration variation on the *X-axis* as the robot advanced with a tripod gait, (**b**) The acceleration variation on the *X-axis* as the robot advanced with a bipod gait.

**Figure 18 micromachines-13-01404-f018:**
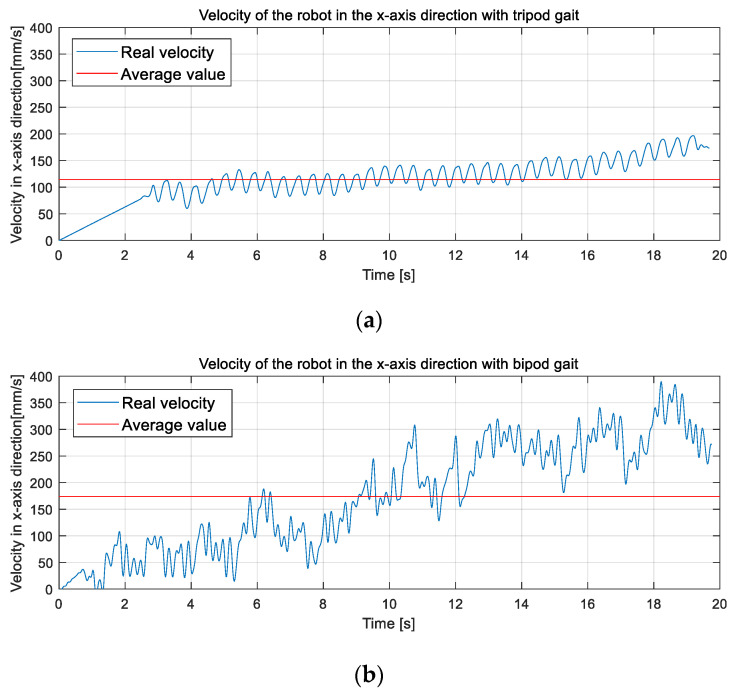
The experimental result of the physical prototype. (**a**) The velocity variation on the *X-axis* as the robot advanced with a tripod gait. (**b**) The velocity variation on the *X-axis* as the robot advanced with a bipod gait.

**Figure 19 micromachines-13-01404-f019:**
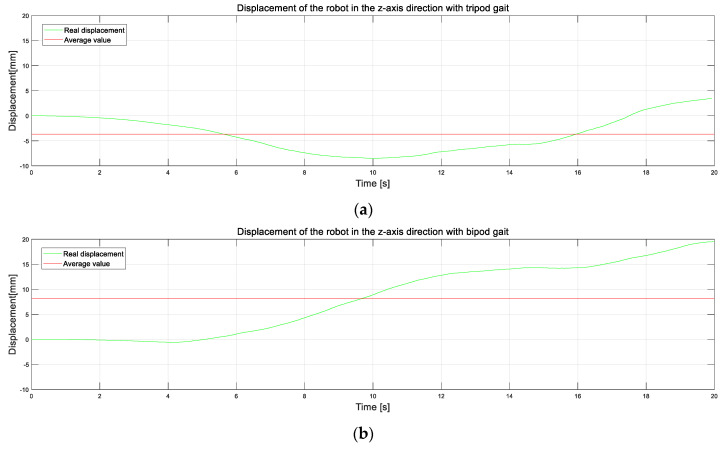
Plot of the actual and average displacements of the robot in the *Z-axis* direction for different gaits. (**a**) The displacement variation on the *Z-axis* when the robot advances with a tripod gait. (**b**) The displacement variation on the *Z-axis* when the robot advances with a bipod gait.

**Table 1 micromachines-13-01404-t001:** Material information of each component.

Component Name	Material	Young’s Modulus (GPa)	Poisson’s Ratio
Drive shaft	C45E4	210	0.31
The crank	AlMg1SiCu	68.9	0.33
U-shaped bar	AlZnMgCu1.5	71	0.33
Other artifacts	Photosensitive Resin	15	0.23

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
