# Peer review of "Design, Analysis and Experiments of Hexapod Robot with Six-Link Legs for High Dynamic Locomotion"

_micromachines, 2022, doi:10.3390/mi13091404_

Round 1

Reviewer 1 Report

Summary:

The authors have designed a five-bar linkage to give a robot leg a high stepping range. They built a robot with six of these legs and developed trajectories for walking with two legs on the ground at once and three legs on the ground at once. 

Major comments: 

The authors need a more thorough and accurate survey of the legged robotics literature. As it stands, the literature review barely touches on research from the last 5 years, and misses huge swathes of research that is relevant to the authors' project. 

The authors have clearly done a lot of work to design, simulate, build, and analyze their robot. However, in large part because of the thin literature review that does not provide proper context for the contributions, it is not clear to this reviewer why this robot was built and/or what knowledge has been gained from it. Some example reasons to build a robot are: 

1) For a specific application that no existing robots can yet accomplish

2) To reveal a physical relationship that suggests a design principle for legged locomotors

3) To break a locomotion-related record, such as highest repeatable jump, fastest absolute or relative speed, best stability on uneven terrain

4) To demonstrate a design method that substantially decreases the cost or time required to build a robot

5) To demonstrate a motion or navigation control method

(This is not an exhaustive list.)

What record is this robot breaking? Does the five-bar linkage allow it to clear larger obstacles than other robots of a similar morphology?

Do the authors develop a new method of planning trajectories? What differentiates it from existing methods, and why is it superior? Or are the authors using existing methods to generate the bipod and tripod gaits? (There is no context with which to answer this question.)

What design principle for legged locomotors does this robot reveal? Are the authors testing some of the conclusions from Ramdya et al. with a robot? Have the authors developed a method that other roboticists could use to generate bipod gaits for their robots? Etc. 

Specific comments:

The authors should report not just absolute speeds like 0.5 m/s but also relative speeds, like x bodylengths per second. Also, it is not clear how high 7 mm is relative to the height of the robot. 

On line 27, the authors claim that the two traditional methods of power transmission for legged robots are cable-driven and quasi-direct-drive. These are not the only two methods of power transmission used in legged robots. Many legged robots have a substantial gear reduction and no cables, such as the RHex robot from the Koditschek lab or the Anymal robot from Anybotics and the Hutter lab. Some also use hydraulics, such as Spot and Atlas from Boston Dynamics. The Minitaur robot uses a five-bar linkage and there is a paper from 2016 by Kenneally, De, & Koditschek about the relevant energetics that is not mentioned in the review. Are the authors making the claim that using a five-bar linkage is what distinguishes their legged robot from other legged robots? The authors also do not review any of the other hexapod robots with similar morphologies or mention the many versions of this type of robot that are commercially available. Without this comparison, and with the knowledge that these robots exist, it is impossible to know what this new robot contributes to the literature. Finally, none of the citations given for this claim are from after 2018, and all but one of them are from before 2015. 

In the paragraph that starts on line 33, the authors claim that hexapod robots walk with tripod, quadruped, or pentapod gaits. This phrasing is nonstandard, and appears to have come from Ramdya et al. The meaning of "quadruped" gaits should be explained and citations should be given to support the claims about how hexapod robots typically walk. Typically "bipedal" would refer to the number of legs involved in a gait, not the number of legs on the ground at any given time. For example, a quadruped could walk with a bipedal gait by rearing up onto its hind legs. A trot is typically described as a quadrupedal gait where two legs are on the ground at any given time, not a bipedal gait. The authors do give some explanation starting on line 298 but this comes very late in the paper. 

In line 116, what is "bionics theory"? This should have a citation and an explanation. 

On line 238, the authors say that they are introducing a foot trajectory planning method. Have they developed new theory to do this, or are they applying existing theory? Is this new theory a contribution of the paper? If this is a contribution, the authors need to cite trajectory planning literature and provide context for what exactly they are contributing and what they are applying from existing work. If the application of existing theory is meant to be a contribution, why this is a sufficient contribution for an archival paper rather than a technical report needs to be justified. 

On line 272, I see a couple of issues. First, spiders are not hexapods. Second, if the authors are trying to build a robot that moves like a spider, they need to at the bare minimum cite spider locomotion research and show how their robot moves similarly to how a spider moves. If they did not design the robot with spider locomotion in mind, this claim should be removed. 

In Figure 18, the y axes on the two plots should have the same scale so that they can be compared more easily. The plots should also have titles and since there is only one line on each plot, there is no need for a legend. Alternatively, the authors could combine the two plots into one plot with a title and a legend. 

Reviewer 2 Report

Section 1:

The introduction is not sufficient as it is too broad to mention the related work in the literature and the motivation of the study in this paper is not clear to the reviewer. In addition, the way of citing references is not rigorous.

 Section 2:

The system overview can be improved by: (1) presenting the design diagram (Fig. 2)  with the schematic sketch (Fig. 1); (2) reducing the view used in Fig. 4, in which a maximum of two views are enough; (3) aligning the part 1 at the same height when presenting the lowest and highest position of the leg mechanism, etc

Section 3:

The coordinate system in Fig. 5 is not global, but a local coordinate system of one leg; 

Has the author validated the gait analysis in the experimental tests?

Section 4:

When mentioning the max. lift speed and acceleration, the author should mention what are the inputs of the simulation. 

In table 1, what are the material types, e.g. is it aluminium 7075? 

In Fig 16, y-axis notations are missing. 

Figs. 4, 19, the figures are out of margin.

It is not clear to the reviewer of the results presented in Fig 18.

The reviewer would recommend providing a video as supplement material for this paper.

Round 2

Reviewer 1 Report

Thank you to the authors for your detailed and careful reply. From the changes the authors have made and the explanations in the response, I can see what I was confused about in the initial submission.

The introduction now makes a much more compelling case for what this robot has been built to test and why the contributed leg mechanism is a good option compared to the existing power transmission options available for legged robots. The paper is overall much improved, and the addition of the video is very helpful. 

I do have a few comments based on the revision which I hope the authors will benefit from. 

I was not able to find reference 19 and was therefore unable to determine whether this is the case, but if this is the first robot comparing a tripod to a bipod gait based on the analysis in Ramdya et al. 2017, the authors could make a stronger claim about the contribution from using a robot to test this bipod gait in direct comparison to the more commonly used tripod gait.  This is a contribution that is useful not just for robots with the specific morphology of a round-ish hexapod, but might also be useful for other morphologies of legged robots, such as rectangular hexapods like RHex-family robots or potentially even quadrupedal robots like the Stanford Doggo if the reader can do a little work to figure out how to apply this insight. As a reader who has developed gaits for existing robots with both 4 and 6 legs to adapt them for new tasks, this kind of information about the stability-speed trade-off would have been very interesting to me when I was doing my initial literature review to come up with ideas and see what other people had already discovered on the topic. Giving a little more emphasis to the contribution of comparing the two gaits earlier on in the paper might help expand your readership to include people who are adapting existing robots for new tasks as well as people who are interested in building new robots using your leg mechanism. 

I was also able to catch a couple of typos that I missed in the initial round which could improve understanding for the reader.

In Fig. 18, the titles for the two plots are the same. The (b) plot should have "bipod" in the title instead of "tripod."

Fig. 19 should also have "tripod" and "bipod" mentioned in the titles of the plots. 

Reviewer 2 Report

The introduction needs further improvements by a thorough survey of (1) existing typical legged robots (particularly in Hexapods) with linkage-based leg designs, (2) typical gaits of Hexapods with particular focus on related gaits used in this work;

In Section 2.1, why link 6 in Fig. 1a is missing when describing the leg mechanism? Shouldn't it be a six-link leg? Furthermore, from a linkage point of view, this linkage looks to the reviewer more like a 4-bar linkage (loop 1-2-3-4-1), due to which the DoF is only 1.

In Fi.g 15, add the axis notation with units for three subfigures.

In Fig. 16, add the walking direction (X-axis) , Z-axis for a better understanding of the test results in Fig. 19. and annotation with arrows pointing at the key elements, e.g. IMU.

In Fig. 17-19, recommend rearranging the results with two figures: one for tripod gait and the other for biped gait. In addition, the X-axis (Time) should be synchronized. 

Round 3

Reviewer 2 Report

Thanks authors to address the comments and there are two minor following comments that the reviewer would make:

In Fig. 15, there is no axis notation with units for the Y-axis, due to which the read will have no idea of the displacement, speed, and acceleration.

Since the authors are mentioning different gaits for the hexapods (roundish or rectangular), there is one recent work the reviewer should be aware of, it studied different gaits for a hexapod robot due to its reconfigurability.

[Ref] J. Camacho, M. Wang, M. Russo, X. Dong, D. Axinte. "Novel reconfigurable parallel kinematic walking machine tool enables symmetric and non-symmetric gait configurations",

IEEE/ASME T-MECH, 2022. https://doi.org/10.1109/TMECH.2022.3183689